# Gender Differences in Caring for Children with Genetic or Rare Diseases: A Mixed-Methods Study

**DOI:** 10.3390/children9050627

**Published:** 2022-04-27

**Authors:** Shao-Yin Chu, Chin-Chen Wen, Chun-Ying Weng

**Affiliations:** 1Genetic Counseling Center, Buddhist Tzu Chi General Hospital, 707, Section 3, Chung-Yang Road, Hualien 97074, Taiwan; chushaoyin@gmail.com (S.-Y.C.); cyweng@tzuchi.com.tw (C.-Y.W.); 2Department of Pediatrics, Buddhist Tzu Chi General Hospital, 707, Section 3, Chung-Yang Road, Hualien 97074, Taiwan; 3Department of Medicine, College of Medicine, Tzu Chi University, 701, Section 3, Chung-Yang Road, Hualien 97074, Taiwan; 4Department of Human Development and Psychology, College of Humanities and Social Sciences, Tzu Chi University, No. 67, Jieren St., Hualien 97074, Taiwan

**Keywords:** family caregivers, gender differences, genetic or rare diseases, health outcomes, illness perception, parenting stress

## Abstract

As a factor in parenting stress, gender differences in caring for children with genetic or rare diseases warrant research attention; therefore, this study explored gender differences in parenting stress, health outcomes, and illness perceptions among caregivers of pediatric genetic or rare disease populations to improve the understanding of such gender differences. Applying a concurrent triangulation mixed-methods design, we conducted a questionnaire survey to assess study measures for 100 family caregivers (42 men and 58 women), which included a free-text response item to probe caregivers’ subjective perceptions of the children’s illness. The gender differences hypothesis was tested with statistics and the qualitative data about illness perception was analyzed by directed content analysis. Most female caregivers served as the primary caregivers and provided more caregiving, while they experienced significantly increased levels of parenting stress and depressive symptoms compared with male caregivers. Female caregivers perceived the conditions of their children’s diseases to be highly symptomatic, with negative consequences and requiring disease control. By contrast, male caregivers had stronger perceptions regarding the negative effects of the disease on the children’s quality of life. The gender discrepancy in viewpoints of illness perception sequence may contribute to female caregivers’ higher levels of stress and depressive symptoms than males.

## 1. Introduction

Genetic diseases are a group of disorders caused by complete or partial abnormal change in DNA sequences, one or multiple gene mutations, a combination of gene mutations and environmental factors, or chromosomal damage [1]. Rare diseases, by contrast, are characterized by their infrequent occurrence and are differently defined across countries (e.g., as diseases with a prevalence of less than 1 in 200,000 in the United States; of 5 in 10,000 in European countries; and of 1 in 10,000 in Taiwan) [2,3,4]. Most rare diseases are genetic disorders [5]. Because of the inheritability of genetic and rare diseases, these disorders, such as congenital hypothyroidism and congenital adrenal hyperplasia, may present in an individual prenatally or after birth, and can be detected through prenatal genetic examination or newborn screening with appropriate medical care [6,7,8]. However, some cases of genetic or rare diseases may have gradual onset, delayed diagnosis, receive no effective treatment, or entail chronically comorbid conditions, which can lead to delayed development, intellectual disability, and maladaptive emotional or behavioral responses or can become life threatening. This can affect the quality of life of the children with such diseases and their families [9,10,11]. Studies have indicated that family caregivers who serve as the primary caregivers face higher medical care burdens and are often psychosocially or financially affected by this role [12,13]. However, such caregivers generally receive little information or support from health-care providers [14]. Family caregivers also experience emotional burdens, such as feelings of isolation or illness-related distress [15,16], and perceive themselves to be stigmatized in academic, medical, community, and family settings [17].

However, few studies have investigated how gender influences caregiving for children with genetic or rare diseases. According to traditional gender roles, women are socially and morally expected to serve as primary informal care providers for family members with illnesses [18], and consequently face higher levels of caregiving burdens and psychological distress [19]. A quantitative study revealed that such differences in gender expectations are present when caring for children with rare diseases, with mothers being reported to experience higher levels of parenting stress and distress than fathers [20]. Another quantitative study found that the quality of life of mothers was significantly lower than that of fathers after a child with a disability obtained a genetic diagnosis [9]. Another qualitative interview study revealed that, although both parents’ lives were influenced by their child’s diagnosis to a similar degree, cultural expectations for gender roles led to some caregiving tasks being viewed as gender specific [21]. For example, fathers reportedly became more focused on their jobs and financial matters, whereas mothers often left or changed their jobs to focus on caring for their child’s everyday needs and to maintain their family relationships.

Although studies have reported gender differences with respect to parenting stress and caregiving tasks, how gender affects illness perception is unknown. Illness perception involves an individual’s knowledge and beliefs about symptoms, illnesses, medical conditions, and health systems [22]. It also involves individuals’ subjective understanding or lay beliefs regarding illness and health, which are entrenched in the self and in sociocultural systems. Leventhal, Brissette, and Leventhal proposed that lay people’s thoughts on the threat of an illness could be organized into five dimensions: identity, timeline, causes, consequences, and cure and control [23]. According to Leventhal’s common-sense model (CSM) of self-regulation, each of an individual’s cognitive representations guides their selection of coping behaviors for controlling an illness, which in turn can determine their health or emotional outcomes [24]. Because illness occurs within the family, every component of the self-regulation process can affect family members [25]. For example, perceptions of the severity of the consequences and the chronicity of a child’s disease were reported to be associated with parental depressive symptoms [26] and caregiver burdens [27]. In addition, discrepancies in illness perceptions among family members were reported, with one study indicating that mothers focused more on the negative aspects of their children’s chronic illnesses than fathers did [28].

This study explored gender differences in parenting stress, health outcomes, and illness perceptions among family caregivers of children with genetic or rare diseases. Using a concurrent triangulation mixed-methods design [29,30], this study converged both quantitative and qualitative data to enable a holistic evaluation of the gender differences in caregiving. A questionnaire survey was designed to assess caregiving tasks, parenting stress, and health outcomes according to Lazarus’s view on stress [31]. In addition, a free-text response item was included to inquire about family caregivers’ perceptions of their child’s illness situation according to Leventhal’s CSM [23,24]. This study hypothesized that male and female family caregivers would differ significantly with respect to caregiving tasks, parenting stress, and health outcomes. In addition, this study explored the illness perceptions of family caregivers and how gender discrepancy in illness perception may explain the phenomenon of gender differences.

## 2. Materials and Methods

### 2.1. Participants

Participation in this study was open to the family caregivers of children affected by genetic or rare diseases who attended a regularly scheduled outpatient genetic counseling clinic of a medical center in eastern Taiwan. Family caregivers whose level of Chinese was insufficient to complete a questionnaire survey were excluded. Recruitment announcement was implemented by the first author, a senior attending pediatrician at the clinic, and lasted for one year. A total of 297 attendants diagnosed with 84 types of genetic or rare diseases visited the genetic counseling clinic during the study period. The family caregivers of 99 patients were contacted. Informed consents were finally obtained from 100 family caregivers of 77 patients, of which 50 family caregivers are couples. Each participant was assisted by the third author, a genetic counselor, to ensure the questionnaires were completed after the clinical encounter.

As indicated in Table 1, the 100 family caregivers (42 men and 58 women) were on average 43.4 years old (standard deviation (*SD*) = 11.6), mostly senior high school educated (47%), married (84%), and had served as caregivers for an average of 8.7 years (*SD* = 5.8). Significant gender differences were identified with respect to participants’ occupations (*χ*^2^ [4] = 32.48, *p* < 0.01, Cramer’s *V* = 0.57) and their familial relationships with their children (*χ*^2^ [3] = 100.00, *p* < 0.01, Cramer’s *V* = 1.00). For the male caregivers, 90.5% were employed. For the female caregivers, 50.0% were employed and 41.4% were housewives. In addition, 95.2% of male caregivers were biological fathers and 86.2% of female caregivers were biological mothers.

As presented in Table 2, the 77 children (49 men and 28 women) that the caregivers were caring for were, on average, 9.9 years old (*SD* = 5.9) and had received a diagnosis (rare diseases, *n* = 33; genetic diseases, *n* = 44) an average of 8.4 years prior (*SD* = 5.6). Few patients demonstrated high heterogeneity in their diagnoses (16 rare disease types and 16 genetic disease types). These diseases were widely categorized into 10 disorders (e.g., congenital metabolic disorders, brain/nervous system disorders, kidney and urinary system disorders, skin disorders, muscle disorders, bone and cartilage disorders, endocrine disorders, congenital malformation syndromes, chromosomal abnormalities, and other unclassified or unknown causes) according to the ICD-10-CM code.

### 2.2. Design and Procedures

A concurrent triangulation mixed-methods design was used [29,30]. Quantitative and qualitative data were collected at the same time in a questionnaire survey, but were analyzed and presented separately. The study related the qualitative results to the quantitative findings through a discussion to find the overall interpretation of gender differences in caring. The study protocol was approved by the institutional review board of Buddhist Tzu Chi General Hospital (IRB103-127-B).

### 2.3. Materials

Quantitative data. Two open questions were used to evaluate caregivers’ caregiving experiences, namely “How many average hours of care do you provide each day?” and “How many disease-related caregiving tasks do you perform each day (for example, taking the child to the doctor, administering medicine, providing the child with physical exercise, cleaning phlegm, and feeding)?”. Caregivers’ parenting stress was evaluated by the Pediatric Inventory for Parents (PIP), which is a 42-item tool for evaluating the frequency and difficulty of caring events for children with serious illnesses over the preceding week, with items in four domains (communication, medical care, emotional functioning, and role constraints) that are scored using a 5-point scale [32]. The PIP demonstrated strong internal consistency (*α* = 0.80–0.96) and construct validity among parents of children with cancer [32] and is a commonly used assessment of parenting stress among caregivers of children with chronic illness [33].

Caregivers’ physical health was evaluated with a yes–no question: “Have you experienced any serious diseases in the past 2 years?”. Caregivers’ mental health was evaluated by two measures. The Center for Epidemiological Studies Depression Scale Short Form (CES-D Short Form) is a 10-item revised version of the CES-D, a widely used measure to assess depressive symptoms occurring over the preceding week by a 4-point scale in the general population [34]. The CES-D Short Form showed satisfactory internal consistency (*α* = 0.78–0.87) and construct validity [35]. The Satisfaction with Life Scale (SWLS) is a 5-item tool to assess, using a 7-point scale, individuals’ global cognitive judgments of their life satisfaction [36]. The SWLS demonstrated a single factor, and high internal consistency is appropriated for a wide range of groups [37].

Qualitative data. According to Leventhal’s CSM [23,24], an illness representation is guided by current experience with the illness. A free-text response question “What are your illness-related concerns for your child?” was designed to probe the illness perceptions of caregivers in caring their children.

### 2.4. Data Analysis

Quantitative data were analyzed using descriptive statistics to define the caregivers’ demographics, caregiving experiences, and measurement tool variables. In addition, an independent *t* test with Cohen’s *d* for continuous variables and a chi-square test with Cramer’s *V* for categorical variables were used to analyze gender differences and effect sizes. The differences between the groups were considered significant if *p* was smaller than 0.01 or 0.05 (two-tailed). All data were analyzed using SPSS Statistics 20.0 (IBM, Armonk, NY, USA).

Content analysis is a qualitative research technique used to extract desired and specific information from a body of qualitative materials (usually written or transcribed verbal) through the systematical classification procedure of coding and identifying themes or patterns by coders or raters [38]. In addition, qualitative information may be transformed into quantitative information, such as category frequencies or ratings for differentiating experiences or perspectives between individuals or groups [39].

In the present study, the qualitative data were analyzed using a directed content analysis to explore gender discrepancy in the illness perceptions of family caregivers. Directed content analysis is one of the content analysis approaches for which analysis starts with a theory or relevant research findings as guidance for initial coding category [40]. The coding strategies of this study were as follows. First, the male and female text responses were separately coded, and the first and corresponding authors read the text responses and highlighted all instances indicating children’s illness-related concerns (e.g., unable to walk or sit, poor learning ability, and negative impressions from others), determined based on the authors’ impressions. Second, all instances were compared and divided into named topics according to their similarities and differences (e.g., physical development, learning, and stigma). The frequencies of the instances of each topic were separately counted. Finally, the topics and related instances were identified and categorized into the following five dimensions of cognitive illness perception: (1) identity—the symptoms attributed to the illness; (2) timeline—expected duration of the illness; (3) causes—the origin of the illness; (4) consequences—the overall evaluation of the seriousness and impacts of the illness in daily life; and (5) cure and control—the extent to which treatment could cure or control the illness [23]. Discrepancies in the coding were discussed until a consensus was reached.

## 3. Results

### 3.1. Quantitative Results

As presented in Table 1, significant gender differences with medium effect sizes were identified for hours of care per day (*t* (98) = −3.33, *p* < 0.01, *d* = −0.67), the number of caregiving tasks performed per day (*t* (98) = −2.92, *p* < 0.01, *d* = −0.59), levels of parenting stress, cognitively appraised through PIP–total frequency (*t* (98) = −3.34, *p* < 0.01, *d* = −0.68) and PIP–total difficulty (*t* (98) = −2.60, *p* < 0.05, *d* = −0.53), and depressive symptoms (*t* (98) = −3.34, *p* < 0.01, *d* = −0.70). No significant gender differences were observed in life satisfaction and physical health. Female caregivers provided more hours of care and performed more caregiving tasks per day (mean = 12.5 and 4.6, respectively) than male caregivers (mean = 7.0 and 3.2, respectively), and experienced more parenting stress and depression (mean = 94.2 and 76.4, respectively) than male caregivers (mean = 75.8 and 63.8, respectively).

### 3.2. Qualitative Results

Of the 100 participants, two female and six male caregivers indicated no illness-related concerns about their children. However, 92 participants (36 men and 56 women) provided responses with 142 instances of illness-related concerns (men *n* = 51, women *n* = 91; Table 3). All instances were grouped into 15 topics that were further categorized into three dimensions of cognitive illness perception: identity (*N* = 64); consequences (*N* = 58); and control (*N* = 20).

#### 3.2.1. Identity

As presented in Table 3, identity-related cognitive illness perception was most frequently identified by caregivers. The caregiver responses included 64 instances of identity (men *n* = 18, women *n* = 46), which were organized into five topics that were used to represent the main symptoms of the diseases the caregivers’ children experienced: emotional and behavioral problems, physical development, language and communication, eating and weight, and intelligence.

Identity was the most frequently perceived dimension among female caregivers (*n* = 46). Female caregivers described the emotional and behavioral problems of their children as manifesting in various ways (e.g., stubbornness, bad temper, dependence, no patience, irritability, always crying when sick, hyperactivity, sleeplessness, running around and too strong to control, and adolescent emotional instability). Although both male and female caregivers were concerned about delays in their children’s physical and language and communication development (e.g., not tall, unable to walk, unable to speak, and unclear speech), the female caregivers provided more detailed descriptions of the developmental problems (e.g., walking on tiptoes, unequal leg length, osteoporosis, abnormal articulation, and stubbornness). Female caregivers also reported more problems regarding eating (e.g., picky when eating, and overeating) and intellectual deficiencies (e.g., poor memory and no concept of danger or right or wrong).

#### 3.2.2. Consequences

As presented in Table 3, the caregiver responses included 58 instances (men *n* = 25, women *n* = 33) of consequence-related cognitive illness perception, which were organized into seven topics that were used to represent how the diseases affected the caregivers’ children’s lives: good health, illness and medical care, learning, relationships, adaptation, employment, and stigmas.

Consequences was the most frequently perceived dimension among male caregivers (*n* = 25). Male caregivers more likely to perceive the diseases as negatively affecting their children’s good health (e.g., health, physical health, physical unhealthiness, physical illness, and normal physical and mental development) and show concern about the illness-related consequences and medical care for their children (e.g., life being threatened when ill, experiencing sleeping difficulties from medical respirators, potential accidents when going out, and experiencing side effects).

In comparison, the female caregivers provided more details regarding their children’s difficulties in forming relationships with their peers (e.g., few friends due to differences in appearance and bad temper, rejection by or wariness from classmates and teachers, verbally or physically bullying peers, or being bullied by peers), and learning problems in school (e.g., educational and learning problems, insufficient learning ability, and requiring postponed enrollment for a year).

#### 3.2.3. Control

As presented in Table 3, the caregiver responses included 20 instances (men *n* = 8, women *n* = 12) of control-related cognitive illness perception. Both types of caregivers indicated that their children had no ability to self-care (e.g., handling menstruation, taking medicine, or self-administering injections) or to live an independent life (*N* = 9). The caregivers further expressed concerns regarding the children’s follow-up medical treatments, education, and long-term care (e.g., child’s placement after the parents grow old) (*N* = 5).

Female caregivers reported more concerns regarding their control as caregivers (*n* = 5), including their economic burdens (e.g., unemployment, obtaining financial subsidies for low-income households, and early interventions) and daily care problems (e.g., feeding, second-hand smoke from the husband’s family members, the mother discovering the child was left unattended at home when she returned to administer the child’s medicine).

## 4. Discussion

This study applied a triangulation mixed-methods design to investigate the gender differences in caretakers caring for children with genetic or rare diseases. The 100 participants (42 men and 58 women, average 43.4 years old, and 84% married) of this study served as family caregivers, having provided care for 77 total children (average 9.9 years old) for an average of 8.7 years. Analysis of the quantitative data revealed significant gender differences in daily caregiving hours and performance of caregiving tasks, parenting stress, and depressive symptoms. These gender differences were clinically significant, with medium effect sizes. As reported in other studies [20,41,42], female caregivers performed more daily care tasks, were more likely to be exposed to parenting stress from communication, medical caregiving, emotional functioning, and role constraints, and were more likely to have recently experienced depressive symptoms. This study also revealed a clear gender-influenced difference in the division of labor in the family, with 90.5% of male caregivers being employed and 50.0% and 41.4% of female caregivers being employed or housewives, respectively. This suggests that women are more likely to adopt the role of stay-at-home parents when their child experiences a chronic medical condition according to traditional gender role expectations in Taiwanese families. It is possible that female caregivers could be what has been termed “lone parents”. Lone parents are those who feel alone when it comes to caring for their child with cancer, regardless of marital/partnership status. In previous published data, lone parents were found to have significantly greater distress. They also reported greater difficulty in meeting the needs of their child with cancer and their other children, and less financial or emotional support than non-lone parents [43,44].

Analysis of the qualitative data revealed that 92 family caregivers had provided responses detailing 142 instances of illness-related concerns regarding their children. Content analysis according to the CSM revealed that both male and female caregivers perceived the same dimensions of their children’s illnesses (i.e., identity, consequence, and control). This indicates that the family caregivers perceived their children’s illnesses to have led to various physical and psychological symptoms, negative impacts on several aspects of the children’s lives, and illness control problems when reflecting on the course of the children’s diseases. This finding was partially congruent with those regarding parental illness perceptions reported in other studies [26,27,45]. The family caregivers of children with genetic or rare diseases perceive the negative consequences of their children’s diseases and their problems with control similarly to how parents of children with mental illnesses do. The caregivers of children with such diseases believed the diseases affected many aspects of their children’s lives, leading to problems regarding the child’s good health, illness and medical care, learning problems, problems regarding peer relationships, stigmatization, and likely problems in future adaptability and employability. The caregivers also indicated that their children had no ability to self-care or to achieve independence, and would rely on lifetime support from caregivers, educational institutions, medical institutions, and long-term care facilities. In this study, we discovered that family caregivers reported the most concern in the identity dimension, with their concerns mostly involving the five main symptoms their children experienced: emotional and behavioral problems, delayed physical development, language and communication problems, intellectual development problems, and eating and weight problems. This may be because perceived dimensions of an illness can differ with the type of disease. Most genetic or rare diseases caused by gene or chromosome mutations affect the brain and nervous (e.g., tuberous sclerosis complex and Rett syndrome), musculoskeletal (e.g., Duchenne muscular dystrophy, mucolipidosis, VACTERL syndrome, and Down syndrome), respiratory (e.g., Rubinstein–Taybi syndrome), and digestive systems (e.g., maple syrup urine disease) [2,4]. Caregivers perceived that multiple body and organ dysfunctions significantly impaired their children’s physical and mental development. This finding is in line with those of other studies [9,10,11].

The quantitative results revealed gender differences in caregiving, parenting stress, and depressive symptoms, and the qualitative results revealed a gender discrepancy in viewpoints about the sequence of three dimensions of cognitive illness perceptions. Female caregivers were more concerned about symptoms than about negative consequences and control, whereas male caregivers were more concerned about negative consequences than about symptoms and control. This suggests that, in female caregivers, perceptions of their children’s illnesses as being highly symptomatic or having a strong illness identity were negatively associated with parenting stress and depressive symptoms, according to Leventhal’s CSM of self-regulation [24]. A possible explanation for this is gendered caregiving influencing the perceived degree of the child’s symptoms. The literature has indicated that mothers of children with rare diseases usually serve as primary caregivers and perform intensive and time-consuming physician-prescribed treatments and home care tasks [46,47]. Mothers are also, generally, the first to recognize symptoms of a disease in children by comparing the child’s behaviors with their caring experiences with other children [48]. However, mothers also often lack information regarding the disease, medical services, and symptom-management strategies [14]. Another explanation may be that the negative affective responses of female caregivers potentially intensify their perceptions of symptoms. Studies have indicated that fathers of children with rare diseases were less emotionally affected by and more accepting of their children’s diseases [47]. By contrast, most mothers had more illness-related distress, including feelings of guilt, worry, sorrow, and anger, long-term uncertainty, and fewer emotional resources [15].

These findings suggest that the psychosocial conditions of female caregivers of children with genetic or rare diseases should be addressed. Psychosocial support should be offered that assists female caregivers in exploring the identity of their children’s diseases, develops their illness perceptions through medical information, offers effective caring and rehabilitation strategies, and teaches caregivers emotional regulation techniques to control their illness-related distress. The female caregivers’ spouses should be invited to consider the gendered discrepancies in caregiving and illness perception related to parenting stress to balance the load of parenting tasks.

The limitations of this study were related to the data collection methods. First, the generalization of the present study results should be approached with caution because of the limited sample size (*N* = 100) recruited from an outpatient genetic counseling center of a medical center in eastern Taiwan. The results regarding the gender segregation of labor in Taiwanese families and gender differences in parenting stress and illness perception might be representative, but cannot be representative of all genetic or rare diseases. The characteristics of rare and genetic diseases with quite low incidence have low prevalence, and are highly heterogeneous. Each disease with different clinical presentations manifests different problems encountered by patients and their caregivers. Secondly, this study applied a free-text response item (What are your illness-related concerns for your child?) to probe caregivers’ illness perceptions regarding their children’s illnesses. Because only a few text responses were obtained, the quantitative content analysis regarding gender discrepancies of illness perceptions and the understanding of caregivers’ representations for their children’s illnesses may have been limited.

Future research may further investigate potential associations between gender discrepancies regarding illness perception and parenting stress and psychological adjustment by administering a questionnaire in larger samples. The illness perception questionnaire is recommended as being specifically applicable to pediatric genetic or rare disease populations [49]. Open or semi-structured interviews guided by Leventhal’s CSM of self-regulation may provide qualitative information regarding gender discrepancies of caregivers’ illness perceptions. Future research should also investigate how family caregivers perceive parenting their ill child, and whether an association exists between the perception of lone parenting and illness perception in Taiwanese families.

## 5. Conclusions

The findings of this study support the existence of gender differences in caregiving, parenting stress, and depressive symptoms in family caregivers of children with genetic or rare diseases. In addition, a gender discrepancy in viewpoints about the sequence of three dimensions of cognitive illness perception was found. Identity may be the key domain of illness perception, leading female caregivers to experience higher levels of parenting stress and depression than male caregivers.

## Figures and Tables

**Table 1 children-09-00627-t001:** Gender comparisons of family caregivers on demographics and measure variables.

Variables	Total (*N* = 100)	Males (*n* = 42)	Females (*n* = 58)	*t* or *χ*2 ^1^	Cohen’s *d* or Cramer’s *V*
Age (y) (mean ± *SD*)	43.4 ± 11.6	44.9 ± 11.0	42.4 ± 12.2	1.09	
Education, *n* (*%*)				3.95	
No school enrollment	2 (2.0)	0 (0.0)	2 (3.4)		
Elementary school	7 (7.0)	3 (7.1)	4 (6.9)		
Junior high school	14 (14.0)	6 (14.3)	8 (13.8)		
Senior high school	47 (47.0)	22 (52.2)	25 (43.1)		
Undergraduate	29 (29.0)	10 (23.8)	19 (32.8)		
Graduate	1 (1.0)	1 (2.4)	0 (0.0)		
Marital status, *n* (*%*)				2.97	
Married	84 (84.0)	37 (88.1)	47 (81.0)		
Partnered	1 (1.0)	1 (1.0)	0 (0.0)		
Single	15 (15.0)	4 (9.5)	11 (20.3)		
Occupation, *n* (*%*)				32.48 **	0.57
Employee	67 (67.0)	38 (90.5)	29 (50.0)		
House husband/Housewife	24 (24.0)	0 (0.0)	24 (41.4)		
Unemployed	4 (4.0)	0 (9.5)	4 (6.9)		
Part-time worker	4 (4.0)	4 (0.0)	0 (0.0)		
Student	1 (1.0)	0 (0.0)	1 (1.7)		
Familial relationship with children, *n* (*%*)				100.00 **	1.00
Parents	90 (90.0)	40 (95.2)	50 (86.2)		
Grandparents	10 (10.0)	2 (4.8)	8 (13.8)		
Years in caregiver role (y) (mean ± *SD*)	8.7 ± 5.8	8.7 ± 5.6	8.8 ± 6.0	−0.15	
Hours of care in per day (h) (mean ± *SD*)	10.2 ± 8.7	7.0 ± 6.9	12.5 ± 9.1	−3.33 **	−0.67
No. of caregiving tasks per day (mean ± *SD*)	4.0 ± 2.5	3.2 ± 2.3	4.6 ± 2.5	−2.92 **	−0.59
PIP Total– Frequency (mean ± *SD*)	86.5 ± 28.6	75.8 ± 23.4	94.2 ± 30.0	−3.34 **	−0.68
Communication	17.2 ± 5.7	14.9 ± 4.5	18.8 ± 6.0	−3.53 **	−0.71
Emotional distress	32.3 ± 11.3	28.5 ± 9.6	35.1 ± 11.7	−3.00 **	−0.61
Medical care	17.7 ± 6.7	15.3 ± 5.6	19.5 ± 7.0	−3.21 **	−0.65
Role constraints	19.4 ± 7.0	17.2 ± 5.6	20.9 ± 7.6	−2.72 **	−0.55
PIP Total–Difficulty (mean ± *SD*)	71.12 ± 24.7	63.8 ± 18.1	76.4 ± 27.4	−2.60 *	−0.53
Communication	14.1 ± 5.1	12.6 ± 3.6	15.3 ± 5.7	−2.72 **	−0.55
Emotional distress	30.0 ± 10.2	25.0 ± 8.0	30.1 ± 11.1	−2.57 *	−0.52
Medical care	12.6 ± 4.8	11.4 ± 3.6	13.0 ± 5.4	−1.69	
Role constraints	16.7 ± 6.5	14.9 ± 4.9	18.0 ± 7.2	−2.43 *	−0.49
Psychological health					
SWLS	20.7 ± 6.2	21.8 ± 5.9	20.0 ± 6.3	1.50	
CES-D (short form)	7.8 ± 6.2	5.4 ± 4.6	9.5 ± 6.7	−3.44 **	−0.70
Recent suffering from illness, *n* (*%*)				0.09	
No	82 (82.0)	47 (81.0)	35 (83.3)		
Yes	18 (18.0)	11 (19.0)	7 (16.7)		

^1^ *t* or *χ*2 denotes the differences of demographics and measure variables between male and female caregivers. *p*-values are for 2-tailed tests. * *p* < 0.05 ** *p* < 0.01

**Table 2 children-09-00627-t002:** Demographic and disease profile of children.

Characteristics of Children (*N* = 77)			
Age (y) (mean ± *SD*)		9.9 ± 5.9	
Years after diagnosed (y) (mean ± *SD*)		8.4 ± 5.6	
Male/female (*n*)		49/28	
Children’s disease conditions			
Rare Diseases, *N* (*%)*	33 (43.0)	Genetic Diseases, *N* (*%)*	44 (57.0)
Diagnosis	*n*	Diagnosis	*n*
Glycogen Storage Disease Type II	1	VACTERL Syndrome	1
Robinow Syndrome	1	Hemifacial Microsomia	1
Tuberous Sclerosis Complex	1	Protein S Deficiency	1
Mucopolysaccharidoses type IIIB	1	Sever Intellectual Disability	1
Mucolipidosis type II	1	Intellectual Disability, R/O Mitochondrial Disease	1
3-Hydroxy-3-Methyl-Glutaric Acidemia	2	R/O Neonatal Intrahepatic Cholestasis caused by Citrin Deficiency	2
Lowe Syndrome	2	Congenital Adrenal Hyperplasia	2
Hereditary Epidermolysis Bullosa	2	Hereditary Blistering Disorder	2
Homozygous Familial Hypercholesterolemia	2	Multiple Disabilities	2
Crouzon Syndrome	2	Chromosomal Abnormality	2
Williams Syndrome	2	Marfan Syndrome	2
Maple Syrup Urine Disease	2	Turner Syndrome	3
Duchenne Muscular Dystrophy	2	Noonan Syndrome	4
WAGR Syndrome	3	Down Syndrome	9
Rubinstein-Taybi Syndrome	4	Congenital Hypothyroidism	10
Prader-Willi Syndrome	5	Congenital Hypothyroidism comorbid with Growth Hormone Deficiency	1

**Table 3 children-09-00627-t003:** Frequencies and percentages of reported instances of each topic and three dimensions of illness perception provided by 92 family caregivers.

Dimension/Topics		Male, *n* (%)	Female, *n* (%)	Total, *N* (%)
Identity	Instances	18	46	64
Emotion/behavior	Stubbornness, bad temper No patience, irritability, always crying, dependent, emotional instability Hyperactivity, running around and too strong to control, disobedient, not sleeping at night, Internet addiction	4 (22.2)	14 (30.4)	18 (28.1)
Physical development	Short stature, delayed physical development, thin, little subcutaneous fat Unable to walk or sit, slow movement, walking on tiptoes Scoliosis, osteoporosis	5 (27.8)	11 (23.9)	16 (25.0)
Language/ communication	No language, slurred speech, impaired articulation Poorly verbalizing their needs, not responding, stubbornness, poor communication skills Delayed linguistic and cognitive development	6 (33.3)	7 (15.2)	13 (20.3)
Eating/weight	Overweight, underweight, poor appetite Poor appetite, picky eating, overeating, enjoyment of eating, choking easily Dietary problems	3 (16.7)	8 (17.4)	11 (17.2)
Intellect	Intellectual disability Poor memory, no concept of danger or left or wrong	0 (00.0)	6 (13.0)	6 (9.4)
Consequences		25	33	58
Good health	Health, physical health, poor physical healthy, physical illness Physical and mental growth	6 (24.0)	4 (12.1)	10 (17.2)
Illness and medical cares	Often ill and requiring clinic visits or hospitalization, experiencing sudden illness, life is threatened when ill Concern regarding accidents when going out, difficulty sleeping due to wearing medical respirator Long-term medicine use, experiencing side effects	6 (24.0)	5 (15.2)	6 (10.3)
Learning	Education, study, and learning problems Insufficient or poor learning ability, unable to keep up with classmates, requiring special education or postponed enrollment for a year	5 (20.0)	6 (18.2)	11 (19.0)
Relationships	Peer relationships, poor peer relationships, no opposite-sex friends, friends are a negative influence Few friends due to differences in appearance and a bad temper, self-consciousness Verbally or physically bullying peers, being bullied by peers Reject or defense by classmates and teachers	2 (8.0)	9 (27.3)	11 (19.0)
Adaptation	Maladaptation to a new school or city Differences in appearance or caring problems after attending school, growing up and handling social perceptions Failed engagement with society, interaction with negative members of society	3 (12.0)	3 (9.1)	6 (10.3)
Employment	Future employment, lack of internship opportunities, early independence and employment, adaptation to employment	2 (8.0)	4 (12.1)	6 (10.3)
Stigma	Negative perceptions or impressions from others Being misunderstood due to behavioral problems of hitting people or scratching and touching things	1 (4.0)	2 (6.1)	3 (5.2)
Control		8	12	20
Children’s control	No ability to self-care, have an independent life, or handle menstruation Noncompliance in taking medicine or receiving injections	5 (62.5)	4 (33.3)	9 (45.0)
Long-term care	Follow-up medical treatments and education Caring problems or placement after parents grow old	2 (25.0)	3 (25.0)	5 (25.0)
Caregivers’ control	Difficulty in life due to caring for three children, unemployment, seeking financial subsidies for low-income households, and seeking early interventions Lack of control in feeding, second-hand smoke from the husband’s family members, mother discovering the child was left at home unattended when she returned to give the child their medicine	1 (12.5)	5 (41.7)	6 (30.0)
**Total instances, *N***		51	91	142

## Data Availability

The data presented in this study are available on request from the corresponding author.

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
