# Peer review of "Gender Differences in Caring for Children with Genetic or Rare Diseases: A Mixed-Methods Study"

_children, 2022, doi:10.3390/children9050627_

Round 1

Reviewer 1 Report

I suggest to improve the qualitative approach description; please insert more details and references about the process. Because content analysis is unique in that it has both a quantitative  and a qualitative methodology 

Author Response

Thank you for this suggestion. Please see the attachment.

Reviewer 2 Report

Thank you very much for your work. It is a manuscript that addresses a problem that deserves to be investigated. However, there are a number of elements that need to be reviewed in depth. Below are the main elements that require further reflection:

  • The description of the instrument needs further substantiation - why was it chosen? What is Cronbach's Alpha? And with the choice of question for the qualitative part of the study as well, what was the reason for selecting that question and not another?
  • The variability of the children's "diseases" is very wide. Would it be possible to encompass it into categories so that the data could be more easily cross-referenced in the results and statistical relationships drawn?
  • The procedure should go in another subsection.
  • There should be more justification as to why such a sample and the size of the sample. At what point is generalization of the results possible?
    Thank you for your time. 

Author Response

(The authors gave the same response as above.)

Reviewer 3 Report

Dear authors:

The manuscript: "Gender differences in caring for children with genetic or rare diseases: A mixed-methods study" is well written and it indicates a great important information.

It is clear and easy to read, and it consideres different points which are of interest for readers, researchers and families. 

I recommend you to include in the methods part a flowchart to understand in a better way the population who was added in the study.

Thank you very much.

Author Response

(The authors gave the same response as above.)

Round 2

Reviewer 2 Report

Thank you very much for your work to improve the manuscript. Another aspect that I would recommend is to review the use of the third person plural. The third person singular should be used to make the work more objective in the writing. Thank you very much for your attention.
